

# Two different microbial communities did not cause differences in occlusion of particulate organic matter in a sandy agricultural soil.

**Frederick Büks[1], Philip Rebensburg[2], Peter Lentzsch[2], Martin Kaupenjohann[1]**

[1] Department of Ecology, Technische Universität Berlin, Germany
[2] Leibniz Center for Agricultural Landscape Research (ZALF e.V.), Müncheberg, Germany



## Abstract

Apart from physico-chemical interactions between soil components, microbial life is assumed to be an important factor of soil structure forming processes. Bacterial exudates, the entanglement by fungal hypae and bacterial pseudomycelia as well as fungal glomalin are supposed to provide the occlusion of particulate organic matter (POM) through aggregation of soil particles.

This work investigates the resilience of POM occlusion in face of different microbial communities under controlled environmental conditions. We hypothesized that the formation of different communities would cause different grades of POM occlusion. For this purpose samples of a sterile sandy agricultural soil were incubated for 76 days in bioreactors. Particles of pyrochar from pine wood were added as POM analogue. One

variant was inoculated with a native soil extract, whereas the control was infected by airborne microbes. A second control soil remained non-incubated. During the incubation, soil samples were taken for taxon-specific qPCR to determine the abundance of Eubacteria, Fungi, Archaea, Acidobacteria, Actinobacteria, α-Proteobacteria and β-Proteobacteria. After the incubation soil aggregates (100-2000µm) were collected by

sieving and disaggregated using ultrasound to subject the released POM to an analysis of organic carbon (OC).

Our results show, that the eubacterial DNA of both incubated variants reached a similar concentration after 51 days. However, the structural composition of the two communities was completely different. The soil-born variant was dominated by Acidobacteria,

Actinobacteria and an additional fungal population, whereas the air-born variant mainly contained β-Proteobacteria. Both variants showed a strong occlusion of POM into aggregates during the incubation. Yet, despite the different population structure, there were only marginal differences in the release of POM along with the successive destruction of soil aggregates by ultrasonication. This leads to the tentative assumption

that POM occlusion in agricultural soils could be resilient in face of changing microbial communities.




# 1 Introduction

Microbial communities play an irreplaceable role in soil ecosystems. Due to their metabolic diversity and abundance, especially bacteria and Fungi have considerable influence on mineral and organic matter transformation (Torsvik and Øvreås, 2002; Gianfreda and Rao, 2004; Uroz et al., 2009; Madigan et al., 2015) and often represent the first element in manifold faunal food webs. They also release a broad variety of molecules involved in nutritional or functional cell-plant symbioses supporting plant growth and health (Pühler et al., 2004; Van Der Heijden et al., 2008).

This work focus on a further ecological function of microbial communities: There is evidence, that the soil microbial community takes part in soil aggregate formation, which is supposed to be positively related to the occlusion of particulate organic matter (POM) within soil aggregates. The grade of occlusion influences the carbon cycle, as occluded POM is superior protected against microbial degradation compared to free POM and mutually promotes development of stable macroaggregates. (Jastrow and Miller, 1997; Bronick and Lal, 2005; Brodowski et al., 2006; Lützow et al., 2006)

The physico-chemical mechanisms underlying aggregate formation comprise interactions between permanent and variable charges of silicates, (hydr)oxides of Fe, Al and Mn, phosphates, carbonates, DOM and POM, which are meditated by multivalent cations with small hydrate shells (e.g. $Ca^{2+}$, $Fe^{3+}$ and $Al^{3+}$), and also hydrophobic interactions (Bronick and Lal, 2005). Fine roots form a physical stabilizing network in and around soil macroaggregates and release cementing root exudates (Bronick and Lal, 2005). The microbial influence is supposed to be achieved by the following mechanisms:

(1) Hyphal Fungi and possibly Actinobacteria as well as filamentous colonies of Cyanobacteria wrap and pervade soil aggregates and increase their mechanical strength (Chenu and Cosentino, 2011). Length, strength, surface adherence and geometry of the mycelia determine the contribution to the bulk stability (Chenu and Cosentino, 2011). When disturbed e.g. by tillage, mycelia were found to be less contributive to the formation of water stable aggregates than intact ones (Beare et al., 1997). Whereas fungal hyphae are assumed to mainly stabilize macroaggregates by formation of a sticky string bag (Gupta and Germida, 1988; Miller and Jastrow, 2000), actinobacterial pseudomycelia were found both within and around soil microaggregates (Kanazawa and Filip, 1986; Ranjard and Richaume, 2001; Mummey and Stahl, 2004).

(2) Microbial exudates and debris adsorb to soil particles and alter their surface properties,



e.g. increase the hydrophobicity which decrease water-caused dispersion of soil aggregates (Chenu and Cosentino, 2011; Achtenhagen et al., 2015).

(3) Microbial biomineralization could cement or block soil particles at their contact regions (Bronick and Lal, 2005). However, little is known about the influence on POM occlusion.

(4) Arbuscular Mycorrhizal Fungi (AMF) are able to produce a proteinaceous substance

opperationally defined as Glomalin Related Soil Fraction (GRSF) or – shortly – glomalin (Wright et al., 1996; Rillig, 2004). It appears in large quantities in various soils (Wright and Upadhyaya, 1998), but is most probably not an exudate, since Driver et al. (2005) showed that >80% of the soil glomalin are strongly bond within hyphal cell walls even after harsh extraction. Soil aggregates rich in glomalin showed a high mechanical stability. However,

the frequently found correlation between soil aggregate stability and glomalin concentration (Rillig et al., 2002; Bedini et al., 2009; Hontoria et al., 2009; Spohn and Giani, 2010; Fokom et al., 2012; Wu et al., 2014) does not necessarily imply glomalin as an agent of soil aggregation, as for example undisturbed AMF populations could produce a lot of glomalin while in effect aggregate soil particles by wrapping. Therefore the influence

of glomalin concentration on POM occlusion is hypothetical.

(5) In contrast to Fungi, the bulk of bacteria is assumed to encapsulate within a viscose matrix of extracellular polymeric substance (EPS) as a reaction to diverse ecological stressors (Roberson and Firestone, 1992; Davey and O'toole, 2000; Mah and O'Toole, 2001; Weitere et al., 2005; Chang et al., 2007; Flemming and Wingender, 2010; Ozturk

and Aslim, 2010). This biofilm contains in average 90% water (Zhang et al., 1998; Schmitt and Flemming, 1999; Pal and Paul, 2008). Only 10% to 50% of the remaining dry mass are microbial biomass, whereas the bulk mainly consists of extracellular macromolecules like polysaccharides, extracellular DNA, proteins, lipids and humic substance (Flemming and Wingender, 2010; More et al., 2014). As a result of the ubiquity (Davey and O'toole,

2000), mechanical strength (viscosity) (Möhle et al., 2007; Flemming and Wingender, 2010), structure (Van Loosdrecht et al., 2002) and distribution across the soil aggregate (Nunan et al., 2003), biofilms are supposed to be an important factor of soil aggregation (Baldock, 2002). However, the viscosity of EPS is affected by its molecular composition (Ayala-Hernández et al., 2008), which strongly depends on species and environmental

conditions: For example, different single-species biofilms cultivated under similar conditions have a strongly differing EPS composition (Béjar et al., 1998; Steinberger and Holden, 2005; Celik et al., 2008). But also similar single-species biofilms show differently composed EPS under varying environmental conditions as demonstrated for





*Pseudomonas aeruginosa* (Marty et al., 1992; Ayala-Hernández et al., 2008). Little is known about the capability of different bacterial taxa to produce EPS. For example, *Rhizibia* species are considered to be strong EPS producers within the phylum of α-Proteobacteria (Rinaudi and Giordano, 2010), and the genetic ability to produce large amounts of high-molecular polysaccharides and proteins was found in different Acidobacteria (Ward et al., 2009). However, these sparse data do not allow predictions about the potential of specific microbial communities to take part in POM occlusion.

The five above specified mechanisms are all supposed to affect POM occlusion, and all of them are obviously influenced by the composition of the soil microbial community. The aim of this work is to test the resilience of POM occlusion in face of the development of two fundamentally different microbial communities in a sandy agricultural soil. In this case study, a gamma-sterilized sandy soil with pyrogenic biochar amendment from pine wood was inoculated in two variants with microbial and sterile soil extract and incubated for 76 days in a bioreactor at field capacity. The second variant was routinely exposed to room air during sampling to initiate the development of an air-born bacterial population. We chose this inoculation, to receive two complex populations that have no potential to converge their taxonomic abundances, as Delmont et al. (2014) recently found that the development of microbial communities is controlled by physical-chemical properties of soils rather than the initial population: E.g. a population taken from a forest soil was given on a sterile grassland soil and there developed like the original grassland population. The biochar was used as a POM analogue, but also represents an upcoming class of soil amendments (Lehmann and Joseph, 2015). During incubation the DNA of Eubacteria, Fungi, Archaea, Acidobacteria, Actinobacteria, α- and β-Proteobacteria in soil samples from both variants was quantified using taxon-specific qPCR. After incubation, soil aggregates of a size between 0.1 and 2 mm were separated by sieving. Following the method of Golchin et al. (1994), aggregates were treated with ultrasound, and the release of intra-aggregate particulate organic carbon (POC) was quantified by use of POM density fractionation and carbon analysis. The amount of released POC depends on the destruction of soil aggregates, which is a function of applied energy, and gives information about the binding strength of POM within the aggregates.

We hypothesized that the establishment of different microbial communities will lead to a different occlusion of POM. A lower occlusion strength is attended by an increased POC release when applying a specific amount of mechanical stress to soil aggregates under further similar conditions.





## 2 Materials and Methods

### 2.1 Preparation of soil and soil extracts

Air-dried soil from a sandy A-horizon (Su3) of an agricultural experimental site in Berge (Germany) was sieved to <2 mm particle size and mechanically disaggregated in a mortar to create an macroaggregate-free soil sample with $C_{org}$=8.7 mg g$^{-1}$ dry soil. The soil sample was amended with 5 vol% of pyrogenic biochar (pine wood, PYREG® GmbH, Dörth/Germany) with a particle size <0.1 mm (71% < 40 μm, see supplements) and

homogenized by end-over end shaking. Subsequently, the biochar-soil-mixture was sterilized with 40.000 Gy using a Cobalt-60 γ-radiation source and an exposure time of 2 weeks following McNamara et al. (2003). The resulting soil had a pH of 7.1 in 0.01M CaCl$_2$ solution, a four times increased $C_{org}$ concentration of 36.2 mg g$^{-1}$ and a grain gross density of 2.54 g cm$^{-3}$.

In addition, 1200 g of untreated fresh soil were extracted with 1560 ml of 10-fold diluted modified R2A broth (0.1 g l$^{-1}$ NH$_4$NO$_3$, 0.05 g l$^{-1}$ yeast extract, 0.05 g l$^{-1}$ soy peptone, 0.05 g l$^{-1}$ casamino acids, 0.05 g l$^{-1}$ glucose, 0.05 g l$^{-1}$ soluble starch, 0.03 g l$^{-1}$ K$_2$HPO$_4$, 0.0024 g l$^{-1}$ MgSO$_4$, pH 7.2 ± 0.2, autoclaved at 121°C for 20 min) (Atlas, 2010) by end-over end shaking for 3 hours. The extract was filtered twice through two layers of laboratory tissue

paper and afterwards split into two halves. One half was autoclaved at 120°C for 20 minutes, whereas the other half remained untreated to provide an inoculum with a soil-born microbial population.

### 2.2 Incubation and sampling

Under sterile conditions, two triplicates of each 300 g sterile soil were filled into pF-bioreactors (Fig. 1) and packed to get a bulk density of 1.36 g/cm³. When closed and connected to a hydrostatic head, the reactors provide constant matrix potential, similar evaporation rates and sterile air supply for soil microbial containment experiments. In the present study, the headspace was continually replaced with a flow rate of 0.4 l min$^{-1}$ by

room air filtered with an 0.2 μm membrane filter. The hydrostatic head was 120 cm (pF 2.08) and thus provided a soil water content of about 35.0 vol% and a soil air content of 11.5 vol%. The water content of 35 vol% equates to 77 ml soil solution. For example, giving 100 ml soil extract to the dry sample, hence 23 ml are subsequently removed by the hydrostatic head when water tension is adjusted. The adjustment of soil water content was

tested in pre-trials with addition of 100 ml of tap water to 300 g of dry soil sample – here



the impounded water was rejected within 15 minutes and the adjustment to 37 vol% soil water content at pF=2 took place within 4 days (data not shown). These characteristics were also assumed for the main experiment.

The first triplicate ($SP_{soil}$) was inoculated with each 100 ml of the non-autoclaved inoculum to reestablish the native microbial population. The sterilized inoculate was given to the second triplicate to start with an abiotic environment, that is susceptible for infection by air-born microorganisms ($SP_{air}$) when exposed to unsterile air. Soil extract exceeding the adjusted soil water content was removed by the hydrostatic head and discarded.

The soil columns were incubated for a total of 76 days. During the incubation, a stress factor setting was established that includes warm-humid conditions from day 1 to 24, warm and drying-out conditions between day 25 and 50 as well as cold-humid conditions from day 51 to 76. This setting is supposed to promote EPS production and fungal growth (Roberson and Firestone, 1992; Di Bonaventura et al., 2008; Borowik and Wyszkowska, 2016). Therefore, incubation took place at room temperature between 24.5°C and 32.5°C until day 50 and at 8°C from day 51 to 76. Hanging water columns were disconnected at day 24 and reconnected after addition of 100 ml of 10-fold diluted modified R2A broth at day 50.

Soil sampling for DNA analysis was performed with sterile plastic pipes used as sampling rings. About 500 mg composite sample compounded of soil from 3 evenly distributed sampling points was taken from each column 18 and 29 hours as well as 3, 5, 16, 49, 51 and 76 days after inoculation. The samples were filled in 2 ml reaction tubes and stored at -20°C for later DNA extraction and quantification. During each sampling the bioreactors with air-born cultures were exposed for 15 minutes to the unsterile room air to enforce infection, whereas the soil-born variant was sampled in a cleanbench. After each sampling, both variants were reconnected to sterile air supply.

After day 76, the soil was removed from the reactors and air-dried for 2 weeks in a laminar flow hood. A pH of 6.8±0.3 was measured for all variants. Afterwards soil aggregates between 0.1 and 2.0 mm in diameter were used for analysis of POM occlusion. In addition, a non-incubated third triplicate ($SP_{control}$) was analyzed in the same way.

## 2.3 DNA extraction and qPCR

DNA was extracted from 370 mg dry soil equivalent by use of a NucleoSpin® Soil Kit (MACHEREY-NAGEL GmbH & Co. KG, Düren/Germany) following the manual instructions. DNA sample purity, represented by 260/230 nm and 260/280 nm extinction











ratios, was determined with a NanoDrop1000 spectrophotometer (NanoDrop Products, Wilmington, DE, USA) and assessed as free of contamination (NanoDrop, 2008).

For quantification of different phylogenetic classes (Acidobacteria, Actinobacteria, α- and β-Proteobacteria) and domains (Archaea, Eubacteria, Fungi), a quantitative real-time PCR with specific primer pairs (Table 1) was performed using a QuantStudio™ 12K Flex Real-

Time PCR System (Life Technologies, Grand Island, NY/USA). The reaction mix per sample contained 4 µl of 5x HOT FIREPol® EvaGreen® HRM Mix ROX (Solis Biodyne, Tartu/Estonia), each 0.25 µl of the proper 10pM fwd and rev primer solution (biomers.net, Ulm, Germany; Table 2), 14.5 µl of PCR $H_2O$ and 1 µl of template DNA solution. Amplification of DNA templates was executed having an initial denaturation at 95°C for 15

min followed by 40 thermocycles consisting of a denaturation at 95°C for 15s, annealing for 20s at primer-specific temperatures listed in Table 1 and elongation at 72°C for 30s. PCR was checked for consistency by melting curve analysis implemented in the QuantStudio™ 12K Flex Real-Time PCR System. Extracted DNA from standard organisms named in Table 1 was used as DNA standard for the relevant taxa, whereas DNA of non-

target organisms from soil samples in return functioned as negative control. Sample-DNA dilution ranged between 1:1 and 1:100 in steps of 1:10.

**2.4 Disaggregation of soil aggregates and quantification of POC**

Successive destruction of soil aggregates by ultrasonication was used to release occluded

POM from its bonding sites (Kaiser and Berhe, 2014). Therefore, in a first step, 75 ml of 1.6 g $cm^{-3}$ dense sodium polytungstate solution (SPT) were added to 15 g of air-dried $SP_{soil}$, $SP_{air}$ and $SP_{control}$ soil samples. After 30 min of SPT infiltration into the soil matrix and centrifugation at 3,569 G for 26 min, the floating free light fraction (fLF) comprising non-occluded POM was collected by filtering the SPT solution through an 1.5 µm pore size

glass fibre filter. In a following step, the remaining soil was filled up to 75 ml SPT solution and ultrasonicated with 50 J $ml^{-1}$ using a sonotrode (Branson© Sonifier 250) to destroy weaker aggregate bonds and release occluded POM. After centrifugation, the floating occluded light fraction ($oLF_{50}$) was collected. For this purpose, the energy output of the sonotrode was determined by measuring the heating rate of water inside a dewar vessel

(Schmidt et al., 1999). Then again the SPT solution was filled up to 75 ml and the sample was treated with an additional energy of 450 J m $l^{-1}$. After centrifugation, the floating occluded light fraction ($oLF_{500}$) and the "sediment", which contains stronger bound POM as well as molecular OM adsorbed to the mineral matrix, were separated and all separated



light fractions (LFs) and sediment samples were frozen at -20°C, lyophilized, ground and analyzed for organic carbon concentration using an Elementar Vario EL III CNS Analyzer. As dissolved organic matter (DOM) were leached by SPT solution during the first step of density fractionation, extracted light fraction OC is interpreted as light fraction POC.

**2.5 Statistical analyses**

The statistical analysis of microbial populations and POC release comprised the calculation of mean values, standard deviations and analysis of variance ($p < 0.05$). After application of the Shapiro-Wilk test (Shapiro and Wilk, 1965) and Levene test (Lim and Loh, 1996) samples were assumed to be normally distributed and to have variance homogeneity. Total bacterial populations were assumed to be similar in a sample, if the absolute difference between the DNA mean values of both variants is smaller than the averaged standard deviation. A repeated measurement design (two-factorial ANOVA) was used to test for significant differences of class, domain and total DNA concentrations and shares between $SP_{soil}$ and $SP_{air}$ within the final period (von Ende, 2001). Particulate organic matter releases of $SP_{soil}$, $SP_{air}$ and $SP_{control}$ were analyzed using one way ANOVA followed by Tukey's test (Christensen, 1996).










## 3 Results

### 3.1 Microbial population analysis

The DNA extracted from both incubated variants shows qualitative differences in the composition of eubacterial populations and further quantitative differences in the fungal population. It is expressed as ng DNA per mg dry soil (ng mg$^{-1}$) and includes intra- and extracellular DNA. (Fig. 2)

The sum of total measured DNA ($DNA_{tot}=DNA_{EUB}+DNA_{FUNG}+DNA_{ARCH}$) in $SP_{air}$ averages 2

ng mg$^{-1}$ until day 6, increases to 13.6 ng mg$^{-1}$ at day 49 and decreases again to 6.8 ng mg$^{-1}$ until day 76. In contrast, $SP_{soil}$ quickly increases from 2.4 ng mg$^{-1}$ at the beginning to 19.6 ng mg$^{-1}$ at day 6 and then decreases to 11.4 ng mg$^{-1}$. Between day 51 and 76 (final period) both variants show a parallel development, but a significant difference in DNA abundance (p=0.049), which is mainly due to fungal DNA. However, both variants have similar total

eubacterial populations ($DNA_{EUB}$, amplified with Eub338/Eub518 primer pair) within the final period with growth curves similar to $DNA_{tot}$. From day 49 to day 76 the population densities of both variants converge. Within the final period their difference fall below the threshold for similarity.

Fungi ($DNA_{FUNG}$) show nearly no growth in $SP_{air}$ and remain at DNA concentrations below

0.2 ng mg$^{-1}$, whereas the fungal population of $SP_{soil}$ grows from 1.11 ng mg$^{-1}$ at day 0 to 5.6 at day 49 and then decreases to 4.7 ng mg$^{-1}$. Fungal populations of $SP_{soil}$ and $SP_{air}$ differ significantly within the final periode (p=0.001). In contrast, the amount of archaeal DNA ($DNA_{ARCH}$) remains <0,002 ng mg$^{-1}$ in both variants and does not show a significant difference.

Some eubacterial classes show significant differences between the variants. The amount of acidobacterial DNA differs significantly within the final period (p=0.003). While $SP_{air}$ does not exceed values of 0.3 ng mg$^{-1}$, the DNA concentration in $SP_{soil}$ increases from 0.4 ng mg$^{-1}$ to values between 2.19 and 3.2 ng mg$^{-1}$. Actinobacteria in $SP_{air}$ exhibit a nearly constant DNA concentration <0.5 ng mg$^{-1}$. In contrast, the $SP_{soil}$ population quickly rises to

1.7 ng mg$^{-1}$ at day 6 and then decreases to 1.0 ng mg$^{-1}$ at day 76. Although $SP_{soil}$ shows an in tendency higher population then $SP_{air}$, differences of both variants within the final period are not significant (p=0.067). The concentration of α-proteobacterial DNA in $SP_{soil}$ quickly rises from 0.1 ng mg$^{-1}$ to 1.0 within 6 days and then decreases continuously to 0.4 ng mg$^{-1}$, whereas $SP_{air}$ does not exceed 0.2 ng mg$^{-1}$. Within the final period there are no significant

differences between the variants (p=0.237). Among the examined eubacterial classes, only



β-Proteobacteria show a significantly higher population in $SP_{air}$ than in $SP_{soil}$: Until day 16 the DNA concentration in $SP_{air}$ remains smaller than 0.1 ng mg$^{-1}$, but increases to 5.9 ng mg$^{-1}$ at the end. In contrast, $SP_{soil}$ quickly increases to 2.8 ng mg$^{-1}$ at day 6 and then stabilizes at around 0.9 ng mg$^{-1}$.

Eubacterial class DNA as a percentage of the total eubacterial DNA (Table 3) shows a dominance of Acidobacteria in $SP_{soil}$ reaching shares of 32.7% (day 51) and 36.8% (day 76), whereas values in $SP_{air}$ stay below 0.9% (p=0.002). Actinobacteria show a 3-fold higher percentage of around 14.6% in $SP_{soil}$ compared to $SP_{air}$ within the final period (p=0.057). In $SP_{air}$ and $SP_{soil}$, α-Proteobacteria show percentages of around 2.4% and 5.2%, respectively, and therefore do not represent a dominant class (p=0.27). In strong contrast, β-Proteobacteria hold increasing percentages of 79.8% and 88.1% in $SP_{air}$ compared to 8.8% and 12.3% in $SP_{soil}$ (p=0.023). Cumulation shows that these classes cover 88.9% to 96.6% in $SP_{air}$, mainly dominated by β-Proteobacteria, and 60.9% to 69.1% in $SP_{soil}$, that is dominated by Acidobacteria, Actinobacteria and also Fungi. In both variants these classes hold an increasing percentage of the total DNA over time.

### 3.2 POC release

The relative light fraction POC release $C_{rel}$ is defined as the ratio of the POC release at the respective energy level ($C_{frac}$) to the cumulative POC release of all collected light fractions plus the sediment ($C_{tot}$): $C_{rel} = C_{frac} \cdot C_{tot}^{-1}$.

$SP_{soil}$ and $SP_{air}$ do not differ in their relative fLF release, which is around 4.6% of the $C_{tot}$. In contrast, the fLF release of $SP_{control}$ amounts to 44.7% (Fig. 3). $SP_{soil}$ releases 2.4% of the $C_{tot}$ within the $oLF_{50}$, whereas $SP_{control}$ releases 10.3% (p=0.051). $SP_{air}$ lies in between releasing 6.3% without a significant difference to both. At 500 J ml$^{-1}$, all variants release similar percentages of $C_{tot}$. The POC release of $SP_{soil}$ and $SP_{air}$ is similar to the amount released at 50 J ml$^{-1}$, whereas $SP_{control}$ is reduced to 1.3%. $SP_{air}$ shows a tendency to exceed $SP_{soil}$ and $SP_{control}$.

The carbon content of each sediment corresponds to the sum of the respective light fraction POC release and amounts to 92.3% (29.9 mg g$^{-1}$) in $SP_{soil}$, 83.9% (26.5 mg g$^{-1}$) in $SP_{air}$ and 43.8 (15.8 mg g$^{-1}$) in $SP_{control}$. Thus, only $SP_{control}$ shows a significantly reduced carbon content remaining in the soil matrix. In consequence, the C-release from $SP_{soil}$ and $SP_{air}$ does not differ significantly in any fraction (although $SP_{air}$ shows a tendency to release more POC than $SP_{soil}$ in both occluded light fractions). In contrast, $SP_{control}$ loses nearly half of its $C_{tot}$ in the fLF and additional 10% after application of 50 J ml$^{-1}$.



## 4 Discussion


The total eubacterial DNA in $SP_{soil}$ und $SP_{air}$ converge between day 6 and day 49 and match the condition for similarity between day 51 and day 76 (the final period). Also the observed eubacterial classes in both variants seem to be established until day 51 and show a stable or slightly decreasing population development within the final period (Fig. 2).

This development lead to a cumulative percentage of Acidobacteria, Actinobacteria, α-Proteobacteria and β-Proteobacteria on total eubacterial DNA, that increases from 88.9% to 96.6% in $SP_{air}$ and from 60.9% to 69.1% in $SP_{soil}$. It can be seen that this bundle of eubacterial classes holds the majority in both variants and becomes increasingly dominant. For these three reasons, the effect of named eubacterial as well as fungal and

archaeal populations on POM occlusion is discussed based on the final period.

Although there is a similar total eubacterial DNA amount, the population structure is strongly varying between the variants: Acidobacteria and β-Proteobacteria show a significant and Actinobacteria an in tendency but not significant difference between variants, whereas α-Proteobacteria, which have low abundances (< 6%) in both variants,

did not develop differently. Beside Eubacteria, a fungal population developed in $SP_{soil}$, whose DNA spans 27.2% to 41.4% of the total measured population ($DNA_{tot}$), whereas only very small amounts of fungal DNA were found in $SP_{air}$ samples. Hence, ecosystems of both variants were dominated by strongly different microbial classes: During the final period Acidobacteria, Actinobacteria and Fungi together hold 61.9% to 71.3% of the total

measured DNA in $SP_{soil}$. In contrast, $SP_{air}$ is strongly dominated by β-Proteobacteria, which provide 79.4% to 87.3% of the total measured DNA. We conclude, that both variants differ in their community structure within the final period. Following our hypothesis, this implies a different POM occlusion in $SP_{soil}$ and $SP_{air}$.

A strong occlusion of POM during incubation becomes apparent comparing the incubated

variants with $SP_{control}$: The carbon content in the fLFs of $SP_{soil}$ and $SP_{air}$ decreased, while increased in the sediment. However, contrary to our hypothesis $SP_{soil}$ and $SP_{air}$ do not show a significant ($p < 0.05$) difference of POM occlusion in any fraction, although $SP_{soil}$ has a tendency to release less POC. Even considering a relation of microbial development and POM occlusion in single parallels, no correlation of the growth of a specific taxon and

POM occlusion was observed (data not shown). The occlusion in both variants is extensive: Total occluded POC amounts to ~30 mg/g dry soil in both variants and therefore exceed occlusion in comparable soils by four-fold (Büks and Kaupenjohann, 2016). Our





POM mainly consists of pyrochar particles <20μm. Since Kaiser and Berhe (2014) reviewed, that microaggregates <63 μm are stable in face of ultrasonication levels >500 J/ml, an occlusion within very stable microaggregates of the sediment is expected. The main biological agent for this occlusion is most likely bacterial EPS (Six et al., 2004). Thus, in the present study POM occlusion exceeds that of a native soil, but is most probably not affected by the community composition.


However, triplicates usually do not provide sufficient test power to avoid type 1 and 2 errors. Therefore the convention of $p<0.05$ only gives a weak statement. If instead discussing the in tendency increased POM occlusion in $SP_{soil}$ as a fact, fungal glomalin and archaeal EPS can be refused as relevant mechanisms: As AMFs are obligatory symbionts of plant roots (Bago and Bécard, 2002), remains of glomalin might exist in the soil sample as a remain from the field, but neither are expected to differ between the variants nor could be enriched by fungal growth. Also archaeal EPS (Fröls, 2013) could be excluded, since Archaea hardly exist in both variants. Low-molecular weight exudates and biomineralization could play a role in physico-chemical POM occlusion, but chemical diversity and unknown effect levels do not allow an estimation of their influence in the present study.



Fungi are highly abundant in $SP_{soil}$. Therefore, wrapping of macroaggregates by fungal hyphae is expected to enhance POM occlusion. In contrast, Actinobacteria, which are assumed to have the capability to form microaggregates, show only slight differences between the variants and are therefore not supposed to contribute to the occlusion of POM. (Aspiras et al., 1971; Gasperi-Mago and Troeh, 1979; Tisdall, 1991; Bossuyt et al., 2001)



As the broad molecular diversity of EPS (Leigh and Coplin, 1992; Votselko et al., 1993; Allison, 1998; Al-Halbouni et al., 2009; Flemming and Wingender, 2010; Ras et al., 2011) develops in dependency of species and environmental factors and affects viscosity, it seems self-evident that two different complex multi-species biofilms should show different binding strength of POM within soil aggregates. However, even assuming no influence of other microbial binding mechanisms, the bacterial community composition seems to be less relevant in the present study. What are the explanations?


First, relicts of the original EPS could endured drying, mechanical dispersion, γ-sterilization and recolonization along the whole soil treatment and form a background load, which overlays the effect of the newly built EPS on POM occlusion. This explanation for similar POM occlusion of $SP_{soil}$ and $SP_{air}$ seems improbable due to γ-degradation and




metabolization (Kitamikado et al., 1990; Wasikiewicz et al., 2005), but cannot finally be ruled out in this work. More likely, the different microbial development in both incubated variants (1) causes only a little difference in EPS molecular composition, that is not sufficient to affect POM occlusion in large extent, or (2) the span of possible molecular EPS compositions has in general no significant influence on the mechanical characteristics of EPS. Furthermore, (3) despite a broad acceptance of EPS as agents of soil aggregation, its influence could be of minor importance under certain conditions (e.g. in sandy soils). (4) Probably, but also not tested, similar POM occlusions in both variants can be caused by a multi-species balancing mechanisms, in which a loss of coherence due to the dominance of one group of taxa is compensated by another group.

Our results only give a first insight to the relation of microbial community composition and POM occlusion. A more quantitative analysis would require more replicate samples, manifold microbial communities and probably soils from different land use. This was beyond the scope of the present study. Our findings show that soil-microbial ecosystems with vastly different community structures can develop a nearly similar grade of POM occlusion. This implies that soil ecosystems could be able to compensate the influence of population shifts on POM occlusion.





## 5 Conclusion

Our incubation experiment demonstrated the possibility to breed stable soil aggregates in the laboratory within 3 month. However, our hypothesis was not supported by the data. After 76 days of incubation, two variants of the same sandy agricultural soil (Su3) established a similar total eubacterial abundance, but different community structures – one strongly dominated by β-Proteobacteria, the other one by Acidobacteria, Actinobacteria and Fungi. Structural differences between these microbial communities did not cause significant differences in the occlusion of POM. This leads to the tentative assumption that POM occlusion in agricultural soils could be resilient in face of changing microbial communities. Nonetheless, a population shift can affect e.g. soil metabolic characteristics. Therefore, the state of the soil microbial community should remain in focus of agricultural practice.







### *Author contribution*

The POM occlusion experiment was designed and carried out by F. Büks, the qPCR by P. Rebensburg. Data were evaluated by F. Büks with contributions from P. Lentzsch. The manuscript was prepared by F. Büks with contributions from M. Kaupenjohann.




## *Data availability*

**Achtenhagen, J. et al.**, 2015, doi:10.1007/s10533-014-0040-9

**Al-Halbouni, D. et al.**, 2009, doi:10.1016/j.watres.2008.10.008

**Allison, D. G.**, 1998, URL: https://tspace.library.utoronto.ca/handle/1807/82

**Aspiras, R. et al.**, 1971, URL: http://journals.lww.com/soilsci/Citation/1971/...

**Atlas, R. M.**, 2010, ISBN: 978-1-4398-0406-3

**Ayala-Hernández, I. et al.**, 2008, doi:10.1016/j.idairyj.2008.06.008

**Bago, B. et al.**, 2002, doi:10.1007/978-3-0348-8117-3_3

**Baldock, J.**, 2002, doi:0-471-60790-8

**Beare, M. et al.**, 1997, doi:10.1016/S0929-1393(96)00142-4

**Bedini, S. et al.**, 2009, doi:10.1016/j.soilbio.2009.04.005

**Béjar, V. et al.**, 1998, doi:10.1016/S0168-1656(98)00024-8

**Borowik, A. and Wyszkowska, J.**, 2016, doi:10.1515/intag-2015-0070

**Bossuyt, H. et al.**, 2001, doi:10.1016/S0929-1393(00)00116-5

**Brodowski, S. et al.**, 2006, doi:10.1111/j.1365-2389.2006.00807.x

**Bronick, C. J. and Lal, R.**, 2005, doi:doi:10.1016/j.geoderma.2004.03.005

**Büks, F. and Kaupenjohann, M.**, 2016, doi:10.5194/soil-2-499-2016

**Celik, G. Y. et al.**, 2008, doi:10.1016/j.carbpol.2007.11.021

**Chang, W.-S. et al.**, 2007, doi:10.1128/JB.00727-07

**Chenu, C. and Cosentino, D.**, 2011, ISBN: 978-1-84593-532-0

**Christensen, R.**, 1996, doi:10.1007/978-1-4757-2477-6_4

**Davey, M. E. and O'toole, G. A.**, 2000, doi:10.1128/MMBR.64.4.847-867.20

**Delmont, T. O. et al.**, 2014, doi:10.1007/s00374-014-0925-8

**Di Bonaventura, G. et al.**, 2008, doi:10.1111/j.1365-2672.2007.03688.x

**Driver, J. D. et al.**, 2005, doi:10.1016/j.soilbio.2004.06.011

**von Ende, C. N.**, 2001, doi:0-19-513187-8

**Fierer, N. et al.**, 2005, doi:10.1128/AEM.71.7.4117-4120.2

**Flemming, H.-C. and Wingender, J.**, 2010, doi:10.1038/nrmicro2415

**Fokom, R. et al.**, 2012, doi:10.1016/j.still.2011.11.004

**Fröls, S.**, 2013, doi:10.1042/BST20120304

**Gasperi-Mago, R. R. and Troeh, F. R.**, 1979, doi:10.2136/sssaj1979.03615995004300040029x

**Gianfreda, L. and Rao, M. A.**, 2004, doi:10.1016/j.enzmictec.2004.05.006

**Golchin, A. et al.**, 1994, doi:10.1071/SR9940285



**Gupta, V. and Germida, J.**, 1988, doi:10.1016/0038-0717(88)90082-X

**Hontoria, C. et al.**, 2009, doi:10.1016/j.soilbio.2009.04.025

**Jastrow, J. et al.**, 1997, doi:0-8493-7441-3

**Kaiser, M. and Berhe, A. A.**, 2014, doi:10.1002/jpln.201300339

**Kanazawa, S. and Filip, Z.**, 1986, doi:10.1007/BF02011205                     545

**Kitamikado, M. et al.**, 1990, URL: http://aem.asm.org/content/56/9/2939.short

**Lane, D.**, 1991, URL: http://ci.nii.ac.jp/naid/10008470323/#cit

**Lehmann, J.and Joseph, S.**, 2015, ISBN: 978-0-415-70415-1

**Leigh, J. A. and Coplin, D. L.**, 1992, URL: http://www.annualreviews.org/doi/pdf/...

**Lim, T.-S. and Loh, W.-Y.**, 1996, doi:10.1016/0167-9473(95)00054-2                     550

**Lueders, T. and Friedrich, M. W.**, 2003, doi:10.1128/AEM.69.1.320-326.2003

**Lützow, M. v. et al.**, 2006, doi:10.1111/j.1365-2389.2006.00809.x

**Madigan, M. et al.**, 2015, ISBN: 978-1-292-06831-2

**Mah, T.-F. C. and O'Toole, G. A.**, 2001, doi:10.1016/S0966-842X(00)01913-2

**Marty, N. et al.**, 1992, doi:10.1111/j.1574-6968.1992.tb05486.x                     555

**McNamara, N. et al.**, 2003, doi:10.1016/S0929-1393(03)00073-8

**Miller, R. et al.**, 2000, doi:10.1007/978-94-017-0776-3_1

**Möhle, R. B. et al.**, 2007, doi:10.1002/bit.21448

**More, T. et al.**, 2014, doi:10.1016/j.jenvman.2014.05.010

**Mummey, D. and Stahl, P.**, 2004, doi:10.1007/s00248-003-1000-4                     560

**Muyzer, G. et al.**, 1993, URL: http://socrates.acadiau.ca/isme/Symposium16/muyzer.PDF

**NanoDrop**, 2008, URL: **http://www.nanodrop.com/Library/nd-1000-v3.8-users-manual**...

**Nunan, N. et al.**, 2003, doi:10.1016/S0168-6496(03)00027-8

**Overmann, J. et al.**, 1999, doi:10.1007/s002030050744                     565

**Ozturk, S. and Aslim, B.**, 2010, doi:10.1007/s11356-009-0233-2

**Pal, A. and Paul, A.**, 2008, doi:10.1007/s12088-008-0006-5

**Pühler, A. et al.**, 2004, doi:10.1016/j.pbi.2004.01.009

**Ranjard, L. and Richaume, A.**, 2001, doi:10.1016/S0923-2508(01)01251-7

**Ras, M. et al.**, 2011, doi:10.1016/j.watres.2010.11.021                     570

**Rillig, M. C.**, 2004, doi:10.4141/S04-003

**Rillig, M. C. et al.**, 2002, doi:10.1023/A:1014483303813

**Rinaudi, L. V. and Giordano, W.**, 2010, doi:10.1111/j.1574-6968.2009.01840.x

**Roberson, E. B. and Firestone, M. K.**, 1992, URL:



http://aem.asm.org/content/58/4/1284.short

**Schmidt, M. et al.**, 1999, doi:10.1046/j.1365-2389.1999.00211.x

**Schmitt, J. and Flemming, H.-C.**, 1999, doi:10.1016/S0273-1223(99)00153-5

**Shapiro, S. S. and Wilk, M. B.**, 1965, doi:10.2307/2333709

**Six, J. et al.**, 2004, doi:10.1016/j.still.2004.03.008

**Spohn, M. and Giani, L.**, 2010, doi:10.1016/j.soilbio.2010.05.015

**Stach, J. E. et al.**, 2003, doi:10.1046/j.1462-2920.2003.00483.x

**Steinberger, R. and Holden, P.**, 2005, doi:10.1128/AEM.71.9.5404-5410.2005

**Tisdall, J.**, 1991, doi:10.1071/SR9910729

**Torsvik, V. and Øvreås, L.**, 2002, doi:10.1016/S1369-5274(02)00324-7

**Uroz, S. et al.**, 2009, doi:10.1016/j.tim.2009.05.004

**Van Der Heijden, M. G. et al.**, 2008, doi:10.1111/j.1461-0248.2007.01139.x

**Van Loosdrecht, M. et al.**, 2002, doi:10.1023/A:1020527020464

**Votselko, S. et al.**, 1993, doi:10.1016/0167-7012(93)90016-B

**Ward, N. L. et al.**, 2009, doi:10.1128/AEM.02294-08

**Wasikiewicz, J. M. et al.**, 2005, doi:10.1016/j.radphyschem.2004.09.021

**Weitere, M. et al.**, 2005, doi:10.1111/j.1462-2920.2005.00851.x

**Wright, S. et al.**, 1996, doi:10.1007/BF00012053

**Wright, S. and Upadhyaya, A.**, 1998, doi:10.1023/A:1004347701584

**Wu, Q.-S. et al.**, 2014, doi:10.1038/srep05823

**Zhang, X. et al.**, 1998, doi:10.1016/S0273-1223(98)00127-9


## Acknowledgement

This project was financially supported by the Leibnitz-Gemeinschaft (SAW Pact for Research, SAW-2012-ATB-3). We are also grateful to our students Kathrein Fischer, Christine Hellerström, Annabelle Kallähne, Paula Nitsch, Susann-Elisabeth Schütze, Anne Timm and Karolin Woitke for pre-trials and soil preparation.



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





## *Tables*

**Table 1:** Target classes and domains, appropriate primer pairs, annealing temperatures (AT) and standard organisms for qPCR. (AWI=Alfred Wegener Institute, Helmholtz Centre for Polar and Marine Research; DSM=German Collection of Microorganisms and Cell Cultures; ZALF=Leibniz Center for Agricultural Landscape Research)

| Target organism | Primer pair | AT | Standard organism (origin) |
|---|---|---|---|
| Archaea | Ar109f / Ar915r | 57°C | *Methanosarcina mazei* (AWI) |
| Acidobacteria | Acido31 / Eub518 | 50°C | *Acidobacterium capsulatum* (DSM11244) |
| Actinobacteria | Actino235 / Eub518 | 60°C | *Streptomyces avermitis* (DSM46492) |
| α-Proteobacteria | Eub338 / Alf685 | 60°C | *Agrobacterium tumefaciens* pGV2260 (ZALF) |
| β-Proteobacteria | Eub338 / Bet680 | 60°C | *Burkholderia phymatum* (DSM17167) |
| Eubacteria | Eub338 / Eub518 | 53°C | *Pseudomonas putida F1* (ZALF) |
| Fungi | ITS1f / 5.8s | 52°C | *Verticillium dahliae* EP806 (ZALF) |



**Table 2:** Applied primer sequences for class- and domain-specific qPCR.

| Primer | Primer sequence | Reference |
|---|---|---|
| 5.8s | 5'–CGCTGCGTTCTTCATCG–3' | (Fierer et al., 2005) |
| Acido31 | 5'–GATCCTGGCTCAGAATC–3' | (Fierer et al., 2005) |
| Actino235 | 5'–CGCGGCCTATCAGCTTGTTG–3' | (Stach et al., 2003) |
| Alf685 | 5'–TCTACGRATTTCACCYCTAC–3' | (Lane, 1991) |
| Ar109f | 5'–ACKGCTCAGTAACACGT–3' | (Lueders and Friedrich, 2003) |
| Ar915r | 5'–GTGCTCCCCCGCCAATTCCT–3' | (Lueders and Friedrich, 2003) |
| Bet680 | 5'–TCACTGCTACACGYG–3' | (Overmann et al., 1999) |
| Eub338 | 5'–ACTCCTACGGGAGGCAGCAG–3' | (Lane, 1991) |
| Eub518 | 5'–ATTACCGCGGCTGCTGG–3' | (Muyzer et al., 1993) |
| ITS1f | 5'–TCCGTAGGTGAACCTGCGG–3' | (Fierer et al., 2005) |




**Table 3:** Measured eubacterial class DNA of SP$_{air}$ and SP$_{soil}$ variant in relation to total eubacterial DNA in percent at days 49, 51 and 76. Within the final period (day 51 to 76) the total eubacterial population is assumed to be similar between both variants. P-values are given for comparison of shares within the final period. (n=3)

| Eubacterial class | SP$_{air}$ | | SP$_{soil}$ | | |
|---|---|---|---|---|---|
| at day | 51 | 76 | 51 | 76 | p-value |
| Acidobacteria | 0.79 | 0.86 | 32.69 | 36.77 | 0.002 |
| Actinobacteria | 5.97 | 5.51 | 14.94 | 14.23 | 0.057 |
| α-Proteobacteria | 2.37 | 2.51 | 4.55 | 5.85 | 0.270 |
| β-Proteobacteria | 79.75 | 88.10 | 8.83 | 12.27 | 0.023 |
| *sum* | *88.88* | *96.57* | *60.88* | *69.12* | |








## Figures

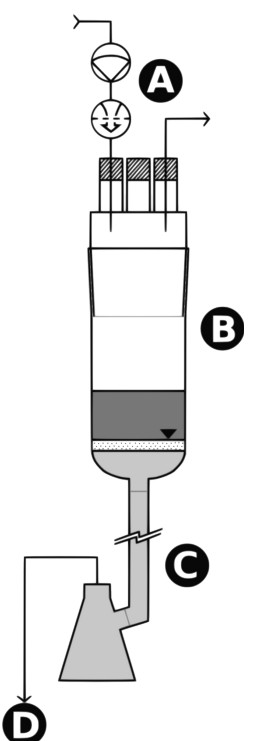







**Fig. 1:** *Field capacity bioreactor with its components A) air supply composed of diaphragm pump and membrane filter, B) filter column with soil sample (dark grey) and filter plate (dotted), C) hydrostatic head (pale grey) and D) liquid waste container.*







**Fig. 2:** DNA concentrations of phylogenetic classes and domains in soil with natural inoculate (SP$_{soil}$) and air-born infection (SP$_{air}$) (values in ng DNA per mg dry soil; * for samples with $p<0.05$; n=3)



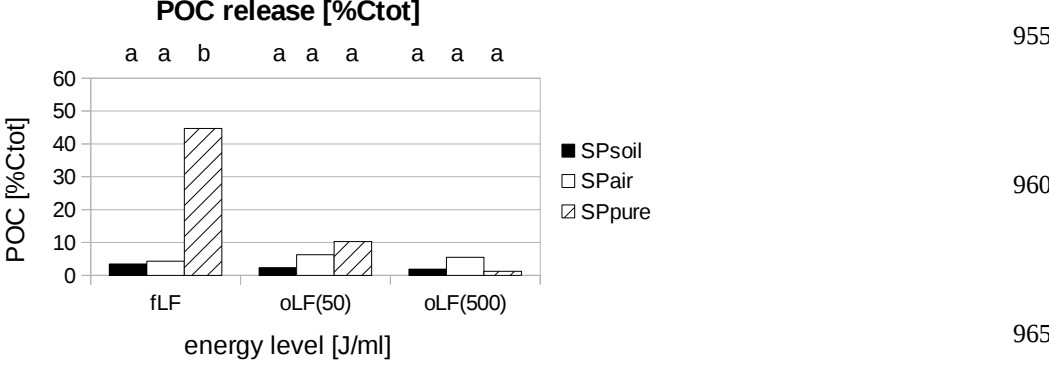

**Fig. 3:** Relative POC release of variants (SP$_{soil}$, SP$_{air}$, SP$_{control}$) at different energy levels (0, 50, 500 J ml$^{-1}$). The highest carbon release is associated with lowest occlusion strength of POM at the respective energy level. (n=3)









