# Peer review of "Two different microbial communities did not cause differences in occlusion of particulate organic matter in a sandy agricultural soil."

_SOIL, 2016_

## Referee Comment (RC1) · Anonymous Referee #1 · 9 Dec 2016

The manuscript presents an experiment to address the level of control soil microbial populations have on the occlusion of organic matter in soil aggregates. Of particular interest is the assessment of how the structure of the different microbial populations might affect the degree of POM capture in aggregates. Unfortunately, as it stands I cannot recommend the manuscript for publication; 1. Statistics: The authors question the statistical power of their own experiment in lines 409 – 410 and specifically state that 'the convention of $p<0.05$ only gives a weak statement' yet throughout the manuscript have used the phrase 'in tendency' (see more in minor comments below) to refer to results where there appears to be some effect of treatment but the p-value is greater than 0.05 and the results therefore cannot be said to be statistically significant.

[Figure]

Having stated that the statistical power of their experiment is possibly too low to make strong statements the authors then go on to discuss one of their weaker results (an increased level of POM occlusion in the SPsoil samples compared to that of the SPair samples) as if it were a strongly significant result. In general the presentation of the results is poor especially in terms of reporting whether the release of POM differed statistically in different soils. Section 3.2 should be the main results section as this is where you test whether the different communities actually make a difference to the amount of POM occluded only one insignificant p-value is noted (line 358) yet when there is a statistically significant result there is no accompanying p-value (line 365). It might be possible to present much of this section as a table to make it clearer. Figure 3 which goes alongside this section has no error bars making it difficult to see if there are differences between treatments within the different fractions. 2. Hypothesis and conclusions: the hypothesis that different microbial communities will lead to different levels of POM occlusion is stated clearly in the abstract (lines41 − 42) and again at the end of the introduction (lines 164 − 165) which is great but this is confused by lines 142 − 144 where the authors state that 'The aim of this work is to test the resilience of POM occlusion in face of the development of two fundamentally different microbial communities in a sandy agricultural soil.' These experiments are not appropriate to test the aim stated in lines 142 − 144, to do this I think one would have to set up replicates of soil with the same microbial communities and test how well POM is occluded, then you would have to change the community in some of the replicates whilst maintaining it in the others and re-test the occlusion of POM to look for differences. This makes the conclusion on lines 478 − 480 speculation. 3. I also have serious concerns about how the manuscript is written, the construction of sentences and paragraphs is poor and because of this it is very often difficult to extract the meaning from whole passages. There is also a great deal of repetition both within and between sections which means the manuscript doesn't flow as well as it could, I think the manuscript requires careful amendment to improve this and hope some of the comments below might help.

Where Comment Line 40 Add the before 'face of different microbial. . .' Line 54 Were

the differences in community structure statistically different? Adding some numbers here – even if it's just a p-value – strengthens the abstract Line 59 Change assumption to conclusion Line 60 By resilient do you mean likely to continue? If you do then I don't think you've actually tested the resilience of POM occlusion in aggregates see comment 2 above. Line 60 Add the before 'face of changing microbial communities.' Line 69 fungi not Fungi, don't really understand the need for the word 'especially' here Line 75 Focuses rather than focus Line 100 Change increase to by increasing and caused to driven Lines 104 – 115 The whole of point 4 in the list is very repetitive, this needs to be re-written to be made more concise. Line 117 Insert 'forming a biofilm; this is' after (EPS), delete as Line 120 on average rather than in average Line 121 Change Only to Between Lines 130 – 134 I found this difficult to follow, this needs to be re-written perhaps something like this? 'biofilm monocultures of different species under similar conditions have divergent EPS composition (Bejar et al., 1998; Steinberger and Holden, 2005; Celik et al., 2008) and monocultures of one species (P. aeruginosa) also give different EPS compositions when exposed to different environmental conditions (Marty et al, 1992; Ayala-Hernandez et al., 2008). Line 139 Change was to has been Lines 142 – 144 As I stated in my longer comments above I don't think these experiments measure resilience and I think this statement confuses what the aim of the manuscript actually is, I think this should be removed. Lines 144 – 163 This is more like materials and methods needs to be removed from the introduction Line 178 Forgive my ignorance but what is a grain gross density? Line 190 Delete each Line 196 Insert (equating to 77ml of soil solution) after 35.0 vol% Lines197 – 199 This whole sentence can be removed if you make the alteration above, I don't really think the example is needed Lines 226 – 229 Move this to the end of line 217, this brings the control treatment into the same paragraph as the experimental treatments and helps the flow of this section. Line 255 Delete 'Therefore, in a first step' and replace with First Line 260 Delete 'In a following step' re-order the rest of this line to read 'Seventy fie ml of SPT solution was added to the remaining soil. . .' Lines 265 – 266 Change sentence to 'Finally, a further 75ml of SPT solution was added to the sample and it was sonicated at 450 J m l-1.' Lines 271 – 272 Rewrite to clarify meaning, perhaps something like this: 'As dissolved organic matter (DOM) was leached by SPT solution during the first density fractionation step the extracted light fraction of OC was interpreted as light fraction POC.' Line 283 Does shares here mean composition? Lines 310 & 312 The authors talk about the DNA extracted on day 6 but in the methods they state DNA was extracted on day 5 Line 320 Add ng mg-1 after 5.6 Line 321 Is the concentration of fungal DNA 4.7ng mg-1 at the end of incubation or on day 51? Line 322 Delete the e on the end of periode Lines 325 – 339 This whole paragraph is difficult to read it feels very repetitive because the authors describe the changes in different populations over time it becomes confusing. I think this needs a substantial re-write for clarity. The authors don't seem to test whether changes in the populations within a treatment are significantly different between different days, this would help describe which populations change. For example the eubacterial populations of SPsoil and SPair become more similar overtime mainly due to changes in the SPair samples, similar results are shown in figure 2 for $\alpha$-proteobacteria and $\beta$-proteobacteria Line 326 Add 'between SP-soil and SPair' after 'differs significantly' Lines 330 – 331 The authors use the phrase 'shows an in tendency' here (and throughout the text) to refer to results where there is a difference but that difference is not statistically significant. Notwithstanding the comments above about whether this is valid given the authors concerns about the statistical power of their experiments if they are going to highlight these results they should use a phrase like ' tends to show a' higher/ lower etc. Lines 353 – 355 This is more of a methods statement, should be removed from the results section Lines 356 – 357 If you re-write these two sentences you can make a positive statement about a significant result. For example: 'The presence of a microbial community contributes significantly (put a p-value here) to the occlusion of POM, in control soils the fLF accounts for 44.7% of the total C whereas in SPsoil or SPair samples the fLF accounts for approximately 4.6% of the total C.' Lines 359 – 360 'At 500 J ml-1, all variants release similar percentages of Ctot.' There is no information on how much this actually is. Figure 3 Error bars? Also by SPpure in the legend do you mean SPcontrol? Line 371 And not und

Line 372 You've defined days 51 to 76 as the final period several times now I don't think you need to do it again Line 380 Are not is Line 382 Replace 'strongly varying' with significantly different Line 383 In tendency statement again – the actinobacteria populations in SPsoil and SPair are significantly different at the beginning of incubation but they seem to converge towards the final period. I think there needs to be some analysis of how the populations change overtime within treatments Line 385 Replace Beside with Alongside Line 392 – 393 'Following our hypothesis, this implies a different POM occlusion in SPsoil and SPair.' I think this is a statement too far, all the DNA profiles tell us is that the treatments were successful in establishing different microbial communities, it doesn't tell us anything about POM occlusion. Line 402 By comparable soils do you mean the control treatment? Lines 409 – 419 Other than my comments above about the statistics this paragraph is very confusing. Lines 410 – 412 I find particularly difficult. Lines 420 – 425 Why could the actinobacteria not be contributing to POM in both SPsoil and SPair? The occlusion is not significantly different between the two treatments so why would the increased levels of fungi in SPsoil enhance occlusion? Line 428 Do you mean independently of or dependent upon? Line 432 Replace 'What are the explanations?' with Possible explanations include: Line 433 Relics rather than relicts Line 434 Replace along with throughout Lines 433 & 436 Replace y in y-radiation with $\gamma$ Lines 472 – 473 I think this statement is a stretch, the experiment didn't specifically test for aggregate stability it tested to see how much POM was released at different energy levels. This either has to be removed or re-written. Lines 478 – 480 The experiment doesn't test resilience this has to be removed as per my comments above. Table 3 Don't understand why the total DNA from days 49, 51, and 76 was used and then only compared to the compositions of days 51 and 76. Surely this should be something like acidobacterial DNA day 51 / total eubacterial DNA day 51 if the aim is to compare the community composition across treatments between days. Figure 2 In the caption replace '* for samples with p<0.05; n=3' with * denotes populations that are significantly different between SPsoil and SPair p<0.05; n=3

---

## Referee Comment (RC2) · Anonymous Referee #2 · 15 Dec 2016

In the draft untitled "Two different microbial communities did not cause differences in occlusion of particulate organic matter in a sandy agricultural soil", the authors compared the occlusion in aggregates of newly added pyrochar particles in the presence of two differing microbial communities. The experiment was conducted in laboratory conditions using a sandy soil. The draft is clear but I have several strong concerns about the methodology and the rationale of this work.

It remains unclear whether the diversity of microbial communities plays on soil organic matter decomposition (for instance Griffiths et al., 2000; Wertz et al., 2006). To this respect, it seems very unlikely that the diversity of microbial communities can play significantly on soil organic matter occlusion which affects SOM decomposition. The

result of the draft is therefore not surprising.

If we consider that microbial communities can play on the amount of occluded POM, the authors made unappropriated experimental choices if they want to see it. First, as mentioned in the discussion part, occlusion is likely not an important stabilization mechanism in sandy soils. As a result, I can't understand why this study was conducted on a sandy soil and not on a clayed or loamy soil. Second, before playing on SOM occlusion, microbial communities should have influenced soil structure. To my opinion, the first logical step would have been to check whether aggregate-size distributions were different between the two treatments. Third, it has been established that the degradation of fresh particulate organic matter is important for macroaggregate and microaggregate formations (for instance Six et al., 2000). To this respect, the choice of the poorly degradable pyrochar is particularly inappropriate. Indeed, pyrochar is likely a poor C source for microbial activity and the production of microbial-derived binding agents.

In summary, even if it is worth reporting negative results, I consider that this negative result is too obvious to be of interest for SOIL readership.

References: Griffiths et al. (2000) cosystem response of pasture soil communities to fumigation-induced microbial diversity reductions: an examination of the biodiversity–ecosystem function relationship. Oikos 90:279-294.

Six et al. (2000) Soil macroaggregate turnover and microaggregate formation: a mechanism for C sequestration under no-tillage agriculture. Soil Biology and Biochemistry, 32:2099-2103.

Wertz et al. (2006) Maintenance of soil functioning following erosion of microbial diversity. Environmental Microbiology, 8:2162-2169.